# HormoneBayes: A novel Bayesian framework for the analysis of pulsatile hormone dynamics

Margaritis Voliotis[1]*, Ali Abbara[2], Julia K. Prague[2,3,4], Johannes D. Veldhuis[5], Waljit S. Dhillo[2], Krasimira Tsaneva-Atanasova[1]

1 Department of Mathematics and Living Systems Institute, College of Engineering, Mathematics and Physical Sciences, University of Exeter, Exeter, United Kingdom, 2 Department of Metabolism, Digestion and Reproduction, Imperial College London, Hammersmith Hospital, London, United Kingdom, 3 Department of Diabetes and Endocrinology, MacLeod Diabetes and Endocrine Centre, Royal Devon and Exeter Hospital, Exeter, United Kingdom, 4 College of Medicine and Health, University of Exeter, Exeter, United Kingdom, 5 Emeritus Mayo Clinic, Rochester, Michigan, United States of America

* m.voliotis@exeter.ac.uk

**Data Availability Statement:** Code can be downloaded from https://git.exeter.ac.uk/mv286/hormonebayes.

## Abstract

The hypothalamus is the central regulator of reproductive hormone secretion. Pulsatile secretion of gonadotropin releasing hormone (GnRH) is fundamental to physiological stimulation of the pituitary gland to release luteinizing hormone (LH) and follicle stimulating hormone (FSH). Furthermore, GnRH pulsatility is altered in common reproductive disorders such as polycystic ovary syndrome (PCOS) and hypothalamic amenorrhea (HA). LH is measured routinely in clinical practice using an automated chemiluminescent immunoassay method and is the gold standard surrogate marker of GnRH. LH can be measured at frequent intervals (e.g., 10 minutely) to assess GnRH/LH pulsatility. However, this is rarely done in clinical practice because it is resource intensive, and there is no open-access, graphical interface software for computational analysis of the LH data available to clinicians. Here we present *hormoneBayes*, a novel open-access Bayesian framework that can be easily applied to reliably analyze serial LH measurements to assess LH pulsatility. The framework utilizes parsimonious models to simulate hypothalamic signals that drive LH dynamics, together with state-of-the-art (sequential) Monte-Carlo methods to infer key parameters and latent hypothalamic dynamics. We show that this method provides estimates for key pulse parameters including inter-pulse interval, secretion and clearance rates and identifies LH pulses in line with the widely used deconvolution method. We show that these parameters can distinguish LH pulsatility in different clinical contexts including in reproductive health and disease in men and women (e.g., healthy men, healthy women before and after menopause, women with HA or PCOS). A further advantage of *hormoneBayes* is that our mathematical approach provides a quantified estimation of uncertainty. Our framework will complement methods enabling real-time *in-vivo* hormone monitoring and therefore has the potential to assist translation of personalized, data-driven, clinical care of patients presenting with conditions of reproductive hormone dysfunction.

**Funding:** MV and KTA acknowledge the financial support of the EPSRC via grants EP/T017856/1 and EP/N014391/1, and BBSRC via grants BB/S000550/1 and BB/S001255/1. JKP is supported by a NIHR academic fellowship, MRC (MR/M024954/1), and Expanding Excellence in England (E3) - Exeter Diabetes Research Unit. AA is supported by an NIHR Clinician Scientist Award (CS-2018-18-ST2-002). WSD is supported by an NIHR Senior Investigator Award. The funders had no role in study design, data collection and analysis, decision to publish, or preparation of the manuscript.

**Competing interests:** The authors have declared that no competing interests exist.

## Author summary

Pulsatile hormone secretion is a widespread phenomenon underlying normal physiology and is also disrupted in many common endocrine disorders. To aid assessment and quantification of hormonal pulsatility, we developed *hormoneBayes*, a novel open-access Bayesian framework for analyzing hormonal measurements. The framework uses mathematical models to describe pulsatile dynamics, together with Bayesian methods to infer model parameter from data. We demonstrate HormoneBayes utility by analysing pulsatility of luteinising hormone (LH) data in different clinical contexts including in reproductive health and disease. Our framework in combination with real-time *in-vivo* hormone monitoring has the potential to assist translation of personalized, data-driven, clinical care of patients presenting endocrine disorders.

## Introduction

Pulsatile hormone dynamics are ubiquitous and play a crucial role in the regulation of many bodily functions related to metabolism, stress, and fertility [1,2]. Hormones are typically secreted in both a basal manner to maintain steady state levels, as well as with superimposed interspersed transient bursts (pulses) [3]. It is now established that the pulsatile nature of hormonal secretion affects their interaction with receptors and downstream effector action [4–6].

With regards to fertility, the hypothalamus is the central regulator of the reproductive endocrine axis. Notably, gonadotropin releasing hormone (GnRH) is secreted in a pulsatile manner, and seminal studies have demonstrated that this pulsatility is fundamental for its action to stimulate GnRH receptors on pituitary gonadotropes [5]. Moreover, disturbances in GnRH pulsatility are observed in common reproductive disorders including polycystic ovary syndrome (PCOS) in which GnRH pulsatility is increased [7], and hypothalamic amenorrhea (HA) in which GnRH pulsatility is reduced [8]. However, despite this disparate alteration in GnRH pulsatility, differentiation of these two common reproductive disorders, which may both present similarly with menstrual disturbance, can be challenging [9]. LH is measured routinely in clinical practice using an automated chemiluminescent immunoassay method and is the gold standard surrogate marker of GnRH. Furthermore, LH can be measured at frequent intervals (eg 10minutely) to assess GnRH/LH pulsatility, and accurate assessment of LH pulsatility could help facilitate diagnosis and treatment of patients presenting with reproductive endocrine disorders [9]. However, this is rarely done in clinical practice because it is resource-intensive, inconvenient for patients, and there is a lack of software for computational analysis of the LH data readily available to clinicians.

Analysis of hormone pulsatility is a challenging computational problem, primarily due the stochastic nature of hormone dynamics and the consequent pulse-to-pulse variability, but also due to extrinsic factors (such as measurement error) obscuring the observed hormone dynamics [3]. Several computational methods for the analysis of endocrine data have been proposed in the literature [3,10–15], and deconvolution analysis, is the current gold-standard method for analyzing LH pulsatility in humans [3]. However, all methods lack open-access software implementation, with a user-friendly graphical interface that can be readily used by clinicians.

To meet these challenges, we have developed HormoneBayes, a novel, open-access Bayesian framework for the analysis of hormone pulsatility data. HormoneBayes uses a stochastic model, describing hormone levels in the circulation incorporating measurement error, and leverages Bayesian statistics [16] to infer model parameters and latent variable dynamics. We

note that this approach is distinct to the deconvolution-based approach [3,14,15], which employs a single-pulse model to represent the data as a sequence of independent pulses. In this deconvolution-based method, the number of pulses becomes a model parameter that needs to be inferred from the data. In a Bayesian context, this leads to a posterior with unknown dimensions, hence posing significant challenges to inference [17,18]. We show that HormoneBayes can be used to accurately identify LH pulses and estimates clinically relevant measures such as inter-pulse intervals and secretion rates. The framework also provides a handle on estimation uncertainty via Bayesian posterior distributions. We showcase how this feature can be used to enable the understanding of alterations in LH pulsatility by analyzing the effect of direct hypothalamic stimulation using the neuropeptide kisspeptin on a subject-by-subject basis. We also demonstrate that HormoneBayes can be used to analyze LH pulsatility in different clinical contexts/reproductive states (including healthy men, women before and after menopause, and women with reproductive disorders such as HA or PCOS). Importantly, the framework comes with an open-access graphical interface that make the core functionality of the framework easily accessible to clinicians and clinical researchers.

## Results

### Analyzing pulsatile hormone dynamics using the hormoneBayes framework

The hormoneBayes framework allows inference of key physiological parameters describing pulsatile hormone dynamics. The framework utilizes stochastic mathematical models describing circulating hormone levels and state-of-the-art Bayesian machinery to calibrate these models to data of hormone profiles and infer model parameters. **Fig 1** presents a parsimonious model (a simple model with great explanatory power) describing circulating LH levels. The model assumes that there are two modes of LH secretion, namely pulsatile and basal. The

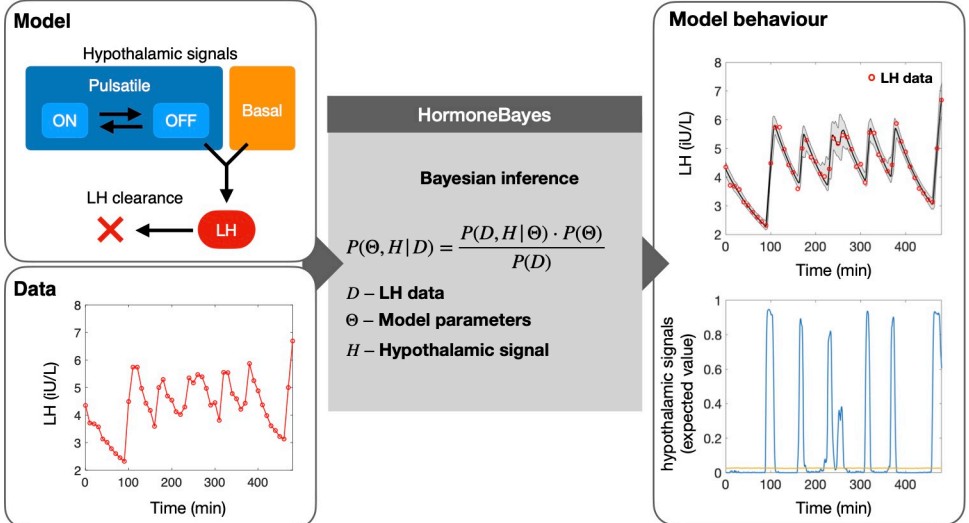

**Fig 1. HormoneBayes: a Bayesian framework for analyzing pulsatile LH dynamics.** The framework uses a parsimonious mathematical model to describe LH levels in circulation as the net effect of secretion and clearance. In the model secretion is driven by a basal hypothalamic signal and a pulsatile signal (mimicking the dynamics of the GnRH pulse generator which can be turned 'on' or 'off'). An efficient Markov-Chain Monte-Carlo (MCMC) method performs the Bayesian inference and extracts model parameters and latent hypothalamic dynamics, which are compatible with the observed data.

former corresponds to an on/off signal that randomly switches between two states (corresponding to a high and a low activity) while the latter corresponds to a continuous noisy signal. Furthermore, the model incorporates LH clearance as a linear first order process leading to an exponential decay of LH levels following a pulse [3]. We note that more detailed clearance models, such as the bi-exponential model [3], could be easily integrated. By considering the processes of LH release and clearance, the model predicts LH circulating dynamics in terms of five key parameters that can be recovered from data: 1. LH clearance rate; 2. maximum LH release rate; 3. strength of the pulsatile signal relative to the basal signal; 4. mean time in the on state; 5. mean time in the off state. Moreover, the model incorporates measurement error as an additional parameter that is determined based on the assay coefficient of variation (CV).

HormoneBayes relies on the Bayesian paradigm to extract information from the observed data. Using the Bayes theorem, the method revises our prior beliefs regarding model parameters by transforming the parameters' prior probability density distributions into posterior distributions. Parameter prior distributions enable the user to input context-specific information into the analysis, hence enhancing the flexibility of the method to handle different datasets. For example, when dealing with data from post-menopausal women the user could choose to adjust the parameter priors to acknowledge a higher LH secretion rate and/or more sustained basal secretion. **Fig 1** illustrates how the Bayesian machinery allows us to calibrate the model to the data and extract information regarding model parameter and latent hypothalamic signals with an estimate of certainty. As we explain in the sections to follow, this information can be used to identify pulses; summarise the data (e.g., providing mean and standard deviation estimates); and perform statistical tests.

## Identifying LH pulses

Using data to infer the latent hypothalamic signal provides a transparent way to identify pulses based on their likelihood under the model. As explained above, the model assumes LH pulses are partly driven by an on/off hypothalamic signal. This latent variable (i.e., inferred variable) takes two values indicating the 'on' (1) and 'off' (0) state of the hypothalamic pulse generator, and therefore the expected posterior estimate (inferred from LH profiling data) can be interpreted as the probability that at any given time the hypothalamic pulse generator is 'on'. This quantitative measure for accessing the likelihood of a pulse can significantly ease the job a clinician trying to decide whether an upstroke in the LH profile represents a pulse or not. **Fig 2**

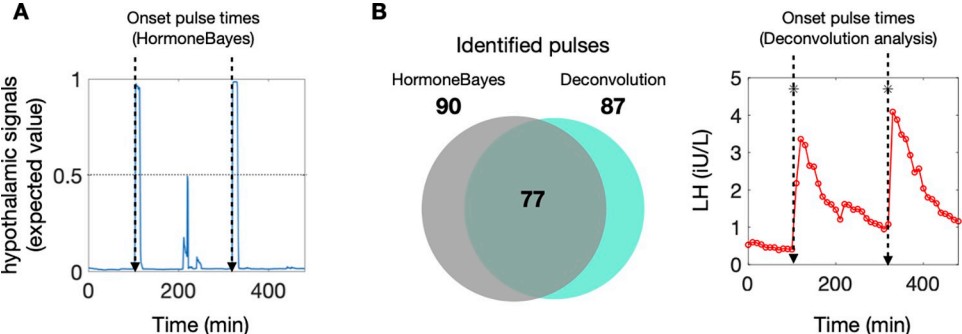

**Fig 2. Pulse identification using HormoneBayes. (A)** Pulses can be identified using the expected value of the pulsatile hypothalamic signal, which can be interpreted as the probability of a pulse at a given timepoint. Here, we mark the onset of a pulse when the pulsatile hypothalamic signal crosses the 0.5 threshold, indicating that at this point a pulse is the most likely event. (B) The majority of the identified pulses (89%, 77/87) are in line with those obtained using the deconvolution method. For the analysis we used LH data obtained from healthy pre-menopausal women in early follicular phase (n = 16).

illustrates a representative example of an LH trace with two obvious pulses occurring at times 100min and 300min. These are indeed identified by inspecting the hypothalamic signal profile which peaks at around those times. At time 200min a less pronounced bump in LH could be indicative of an LH pulse, however the inferred pulsatile hypothalamic signal remains well below 0.5, indicating that under the current model there is higher probability that the bump is a measurement artefact rather than a pulse.

To validate our pulse identification method, we used a database of LH profiles obtained from healthy pre-menopausal women and compared the identified pulses with those previously obtained by the deconvolution method [19], which uses a maximum likelihood approach to fit a series of pulses to the data [3]. Here, we identify a pulse when the posterior probability that the hypothalamic signal is 'on' exceeds a threshold value. We use 0.5 as the threshold value, which signifies there is higher probability that the hypothalamic pulse generator is on (rather than off). This value yields the highest agreement between hormoneBayes and the deconvolution method (see Fig D in S1 Text). We find that hormoneBayes agrees with the deconvolution method in 77 out of 87 identified pulses (89%). Moreover, 13 pulses identified by hormoneBayes were not identified by the deconvolution method. Overall, this suggests a good agreement between the two methods, with hormoneBayes having the added advantage of providing a measure for the likelihood of each pulse that clinicians and researchers can use to inform their clinical decision making.

## Variation of model parameter within and across groups

To test the applicability of hormoneBayes in different contexts we compile a database of LH profiles from four groups with diverse LH dynamics (men, healthy pre-menopausal women with regular menstrual cycles, post-menopausal women, women with HA, and women with PCOS). As illustrated in Fig 3, the model successfully captures the differences in LH dynamics in all four groups. Moreover, the model allows us to summarize LH data through model parameters and assess the variability across and within groups. We find that two model parameters explain most of the variability between groups, namely the maximum secretion rate and pulsatility strength (**Fig 3C**). The first parameter describes how much LH can be secreted over time, whilst the second parameter quantifies the strength of the pulsatile signal relative to the basal signal. Based on these two parameters there is a strong distinction between women with PCOS and women with HA, who have lower LH secretion rates and/or diminished LH pulsatility strength (i.e., pulsatile signal is weak relative to the basal signal). Furthermore, post-menopausal women display higher secretion rates compared to pre-menopausal women but also reduced pulsatility strength (i.e., pulses are less pronounced when higher level of LH are established post menopause). Interestingly, healthy men and women illustrate a much lower parameter variability as compared to HA and post-menopausal women, which could be indicative of various degrees of severity of HA/PCOS and tighter LH regulation in healthy individuals of reproductive age.

## Discussion

We have presented HormoneBayes, a novel computational framework for analyzing hormone pulsatility. The framework combines (i) mathematical (mechanistic) models describing hormone dynamics with (ii) computational Bayesian machinery for inferring model parameters from data. HormoneBayes, comes with an open-access graphical user interface that make the core functionality of the framework easily accessible, a feature lacking from currently available analysis methods.

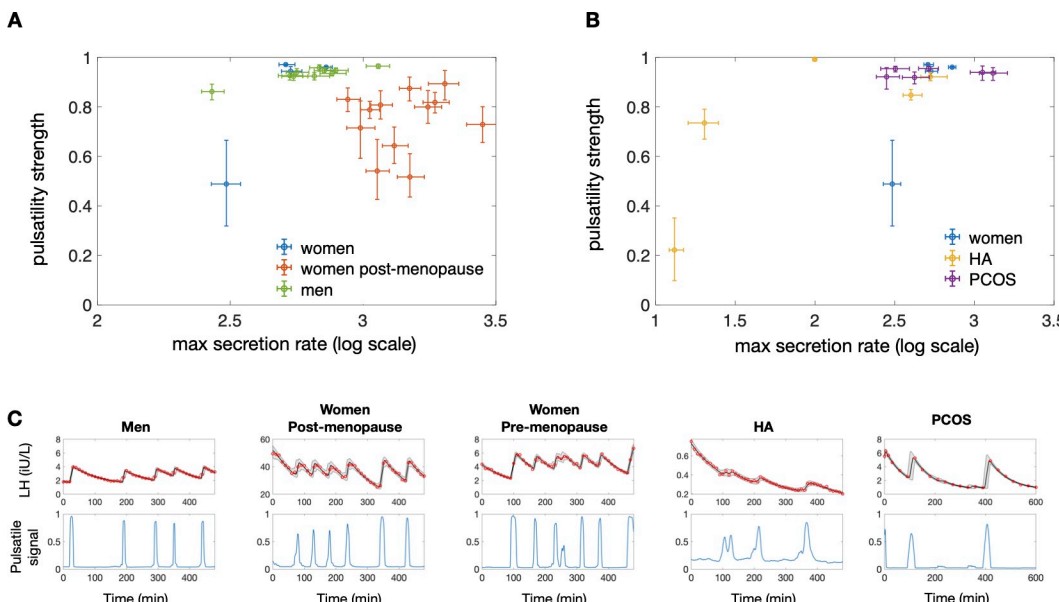

**Fig 3. HormoneBayes handles LH pulsatility analysis in different contexts.** (**A**) Inferred pulsatility strength and maximum secretion rate parameters for different individuals: healthy men (n = 10); healthy post-menopause women (n = 13); healthy pre-menopausal women (n = 4). (**B**) Inferred parameters for healthy pre-menopausal women (n = 4); women with PCOS (n = 6) and women with HA (n = 5) illustrating how the assessment of LH pulsatility could help facilitate diagnosis of patients presenting with reproductive endocrine disorders. (**C**) Representative fits of the model are given for one subject in each dataset.

Using a parsimonious mathematical (generative) model of LH secretion, we have demonstrated the clinical utility of HormoneBayes in accurately describing LH profiles in various contexts (healthy men, healthy pre-menopausal women, post-menopausal women, women with PCOS and women with HA), and for identifying pulses. A novel feature of hormoneBayes is that it summarizes hormone profiles in terms of model parameters that can be used to predict the underlying clinical conditions or reproductive state. Therefore, in the clinic hormoneBayes could assist diagnosis based on hormonal profiles by evaluating how well the profile is described by different model/prior configurations, representing distinct physiological states corresponding to different clinical conditions (e.g., PCOS, HA).

Ultimately, data analysis using HormoneBayes is as credible as the underlying generative model used to describe hormonal dynamics. Unlike deconvolution methods, where the number of pulses is one of the model parameters to be inferred from the data, our approach relies on a generative model that assumes two modes of LH secretion: pulsatile and basal. This assumption is in par with current physiological understanding of the system and the hypothalamic pulse generator hypothesis [20,21]. Furthermore, to model LH circulation levels the model assumes a linear clearance rate. At least one other model used for the analysis of LH pulsatility has used more complex (multiple timescale) clearance dynamics, however we expect this assumption should have a minimal impact for the purpose of assessing LH pulsatility. Nevertheless, the modular design of HormoneBayes allows future extensions of the model with the scope of comparing how well different models capture LH dynamics as well as enabling the analysis of hormone dynamics beyond LH [22,23].

HormoneBayes utilizes the Bayesian paradigm to infer model parameters from the data. Within this paradigm, for each profile the method will output a (posterior) density distribution of model parameters, quantifying how probable parameter values are given the observed profile. This is fundamentally different from non-Bayesian methods, which provide point-

estimates of model parameters, as it allows for statistical testing. For instance, inferred posterior distributions can be used to evaluate the impact of hormonal interventions on LH secretion parameters, and crucially, this statistical evaluation can be conducted not only at the population level but also on an individual basis (see Fig E in S1 Text for an example of this type of personalised analysis). We expect these features of our method regarding personalized analysis will be crucial as measurement technologies mature enabling cheap sampling of hormone levels in real time [22,24].

## Methods

### Ethics statement

Data included in this manuscript were obtained from five different clinical research studies, involving healthy men [25], healthy pre-menopausal women [19], post-menopausal women [26], women with PCOS and women with HA [8]. Ethical approval for these studies was granted by: the Hammersmith and Queen Charlotte's and Chelsea Hospitals Research Ethics Committee (registration number 05/Q0406/142) [8,19]; the UK National Research Ethics Committee-Central London (Research Ethics Committee number 14/LO/1098) [25]; and the West London Regional Ethics Committee (15/LO/1481) [26]. Written informed consent was obtained from all subjects. All studies were conducted according to Good Clinical Practice Guidelines.

### Data collection

Participants attended a clinical research facility for 8 hour study visits hat included baseline (vehicle treatment) LH measurements according to the relevant trial protocol as previously described [8,19,25,26]. A cannula was inserted into a peripheral vein under aseptic conditions (time at least -30 minutes), through which all subsequent blood samples were taken every 10 minutes from time 0 until 480 minutes. All participants were ambulatory and could eat and drink freely during the study visit. All blood samples were left to clot for at least 30 minutes prior to centrifugation at 503 rcf for 10 minutes, after which the serum supernatant was extracted and immediately frozen at -20˚C prior to subsequent analysis using an automated chemiluminescent immunoassay method (Abbott Diagnostics, Maidenhead, UK) in batches after study completion. Reference ranges were as follows: LH 4–14 IU/L; respective intra-assay and inter-assay coefficients of variation were 4.1% and 2.7%; analytical sensitivity was 0.5 IU/L.

### Stochastic model of LH

We used a discrete-time, stochastic model to describe pulsatile LH dynamics. The model comprises of three dynamical variables, $P_t$, $B_t$, and $LH_t$, that describe the pulsatile and basal hypothalamic signals and the LH concertation in circulation, respectively.

The pulsatile hypothalamic signal $P_t$ can take two values: $H_t = 0$ corresponding to the ON state; and $H_t = 1$ corresponding to the OFF state. The stochastic dynamics of $H_t$ are governed by the following probability matrix

|  |  | $H_{s+\delta t}$ | |
|---|---|---|---|
|  |  | **0** | **1** |
| $H_s$ | **0** | $1 - \frac{1}{\tau_{ON}} \cdot \delta t$ | $\frac{1}{\tau_{ON}} \cdot \delta t$ |
|  | **1** | $\frac{1}{\tau_{OFF}} \cdot \delta t$ | $1 - \frac{1}{\tau_{OFF}} \cdot \delta t$ |

i.e., parameters $\tau_{ON}$ and $\tau_{OFF}$ govern the probabilities that the value of $H$ will either flip or remain the same over the time interval $(s, s+\delta t)$.

The evolution of the basal hypothalamic signal, $B_t$, is described using a discrete time autoregressive model obeying the following equation

$$X_{t+\delta t} = X_t - \frac{\delta t}{2} X_t + \sqrt{\delta t} \cdot \varepsilon_t,$$

$$B_t = \frac{1}{1 + e^{-X_t}},$$

where $\varepsilon_t$ is a normally distributed random variable with zero mean and unit variance. Note that both $B_t$ and $H_t$ are bounded in the interval [0,1].

The two hypothalamic signals drive LH secretion, and along with LH clearance dictate the circulating LH levels, $LH_t$. The equation describing the time evolution of $LH_t$, is

$$LH_{t+\delta t} = LH_t + [k(P_t \cdot f + B_t \cdot (1-f)) - d \cdot LH_t] \cdot \delta t$$

where $d$ denotes the clearance rate, $k$ denotes the maximum secretion rate, and parameter $f$ (termed pulsatility strength) describes the relative strength of the two hypothalamic signals. Finally, the model assumes measurement error in the form:

$$LH_t^{obs} = LH_t(1 + \eta_t)$$

where $\eta_t$ is a normally distributed random variable with zero mean and std. deviation equal to the CV of the assay. Throughout our analysis we have used $\delta t = 1$ min, hence, assuming the system dynamics do not change significantly over shorter times.

## Bayesian inference

The hormoneBayes framework uses Bayesian inference to obtain model parameters $\Theta = (\tau_{ON}, \tau_{OFF}, k, d, f)$ and latent variable $(H_t, B_t)$ dynamics from LH profiling data $D$. In particular, hormoneBayes solves the inference problem by sampling from the target posterior distribution:

$$P(\Theta, H_t, B_t | D) = \frac{P(D, H_t, B_t | \Theta) \cdot P(\Theta)}{P(D)},$$

where $P(\Theta)$ is the prior parameter distribution; $P(D, H_t, B_t | \Theta) = P(D | \Theta, H_t, B_t) \cdot P(H_t, B_t)$ is the likelihood of the data given the parameters; and $P(D) = \int P(D, H_t, B_t | \Theta) \cdot P(\Theta)$ is the marginal likelihood or model evidence.

Sampling from the full posterior distribution is performed using a Gibbs sampler, which is an iterative Monte Carlo Markov Chain (MCMC) scheme. The algorithm is initialised with parameter values drawn from the prior distribution, i.e., $\Theta^0 \sim P(\Theta)$ and each subsequent iteration, $i = 1,...,M$ involves two steps: (1) sampling latent variables $(H_t, B_t)^i$ given the data, D, and the current parameter values $\Theta^{i-1}$ and (2) sampling new parameter values $\Theta^i$ given D and the latent variables $(H_t, B_t)^i$. The first step is performed using Sequential Monte Carlo (SMC) with ancestral sampling [27]. The second step is further broken down into two parts, first parameters $(\tau_{ON}, \tau_{OFF})$ are sampled using an adaptive Metropolis-Hastings sampler and then parameters $(k, d, f)$ are sampled using the simplified version of the manifold Metropolis adjusted Langevin algorithm (sMMALA) presented in [28].

For the analysis of all datasets in this study we considered the following independent prior distributions: $\log_{10}\tau_{ON} \sim U(log10(5), log10(240))$ and $\log_{10}\tau_{OFF} \sim U(log10(5), log10(240))$, based on the sampling rate (10min) and duration (480min) used in the LH profiling studies; $\log_{10}(k) \sim \mathcal{N}(0, 5)$, set as a broad uninformative prior; $\frac{\log(2)}{d} \sim \mathcal{N}(80, 9.3)$, based on LH half-

life data; and $\log(f) \sim U(0,1)$, to accommodate analysis of different LH profiles with high or low pulsatility. Evaluation of the algorithm on synthetic dataset can be found in Fig A-C in S1 Text.

HormoneBayes also allows the user to access the effect of pharmacological interventions on LH pulsatility, by fitting in tandem two LH profiling datasets: corresponding to periods before and after the intervention. In this case a composite model is used to allow inference of parameters $\Theta_c = (\tau_{ON}, \tau_{OFF}, k, d, f)$, corresponding to the baseline period (before the intervention), and parameters $\Theta_p = (\tau_{ON,i}, \tau_{OFF,i}, k_i, f_i)$ corresponding to the period after the intervention. Here we assume the intervention does not affect the clearance rate $d$, hence this parameter does not appear in $\Theta_p$. In mathematical terms the target posterior is now given by

$$P\left(\Theta_c, \Theta_p | D_c, D_p\right) = \frac{P(D_c, D_p | \Theta_c, \Theta_p) \cdot P(\Theta_c, \Theta_p)}{P(D_c, D_p)},$$

and sampling from the posterior is performed as described above. An example of this analysis is Fig E of the S1 Text.

An open access C++ implementation of HormoneBayes accompanied with a graphical interface implemented in Python and a user manual can be found at https://git.exeter.ac.uk/mv286/hormonebayes.

## Supporting information

**S1 Text. Supplementary figures.** Fig A: Testing HormoneBayes on synthetic data. Fig B: Assessing the effect of the prior for the LH clearance rate. Fig C: Tuning HormoneBayes when pulses are not clear by using a more informative prior on parameter f. Fig D: Pulse identification using HormoneBayes. Fig E: Using HormoneBayes to identify the effect of interventions on LH pulsatility.
(PDF)

## Author Contributions

**Conceptualization:** Margaritis Voliotis.

**Data curation:** Ali Abbara, Julia K. Prague.

**Formal analysis:** Margaritis Voliotis.

**Investigation:** Margaritis Voliotis, Ali Abbara, Julia K. Prague, Waljit S. Dhillo, Krasimira Tsaneva-Atanasova.

**Methodology:** Margaritis Voliotis, Ali Abbara, Krasimira Tsaneva-Atanasova.

**Resources:** Ali Abbara, Julia K. Prague, Johannes D. Veldhuis, Waljit S. Dhillo.

**Software:** Margaritis Voliotis.

**Writing – original draft:** Margaritis Voliotis.

**Writing – review & editing:** Ali Abbara, Julia K. Prague, Johannes D. Veldhuis, Waljit S. Dhillo, Krasimira Tsaneva-Atanasova.

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
