## [Editor Report · Decision Letter 0]

12 Jul 2023

Dear Dr Voliotis,

Thank you very much for submitting your manuscript "HormoneBayes: a novel Bayesian framework for the analysis of pulsatile hormone dynamics" (PCOMPBIOL-D-23-00942) for consideration at PLOS Computational Biology. As with all papers, your manuscript was reviewed by members of the editorial board. Based on our initial assessment and discussion amongst the editors, we regret that we will not be pursuing this manuscript for publication at PLOS Computational Biology. 

Unfortunately, we feel the tool presented in this manuscript seems to apply to data (dense time courses of lutinizing hormone from patients) that do not typically exist. Therefore the significance is quite limited.

We are sorry that we cannot be more positive on this occasion. We very much appreciate your wish to present your work in one of PLOS's Open Access publications. Thank you for your support, and we hope that you will consider PLOS Computational Biology for other submissions in the future.

Sincerely,

Virginia E. Pitzer, Sc.D.

Section Editor

PLOS Computational Biology

Virginia Pitzer

Section Editor

PLOS Computational Biology

---

## [Decision Letter · Decision Letter 1]

10 Nov 2023

Dear Dr Voliotis,

Thank you very much for submitting your manuscript "HormoneBayes: a novel Bayesian framework for the analysis of pulsatile hormone dynamics" for consideration at PLOS Computational Biology.

As with all papers reviewed by the journal, your manuscript was reviewed by members of the editorial board and by several independent reviewers. In light of the reviews (below this email), we would like to invite the resubmission of a significantly-revised version that takes into account the reviewers' comments.

The reviews noted major deficiencies in model validation/comparison to "gold standard", novelty, true utility for clinicians, and how a personalized analysis would be done. Moreover there seem to be errors in the code implementation noted. These deficiencies would need to be strongly addressed in the revision, in addition to the other major points raised by reviewers.

We cannot make any decision about publication until we have seen the revised manuscript and your response to the reviewers' comments. Your revised manuscript is also likely to be sent to reviewers for further evaluation. We may also try to recruit a third reviewer if necessary to reconcile any remaining points of ambiguity. 

Sincerely,

Marc R Birtwistle, PhD

Academic Editor

PLOS Computational Biology

Virginia Pitzer

Section Editor

PLOS Computational Biology

Reviewer's Responses to Questions

**Comments to the Authors:**

Reviewer #1: This paper describes the use of Bayesian modelling to provide an alternative to the standard deconvolutional methods used to model hormone pulsatility. When applied to the specific hormone LH, these models provide both a close fit to observed data and useful inferences about the underlying hypothalamus-pituitary modulation. The study demonstrates careful use of parsimonious modelling and high-quality human data, and has results suggesting a clear benefit when compared to the use of purely deconvolotional methods. The ability to compare and contrast pre- and post menopausal women is very interesting, as the literature contains conflicting results in this area.

Major comments:

Novelty is an issue. The users should explain how this work differs from PMID: 14601766 and PMID: 29287490 (the latter also providing open-access Baysian code for LH pulsatility).

The results would be improved by (a) an indication of goodness of fit to the data (maybe coefficient of determination added to the upper right subfigure in figure 1) and (b) an example of how a personalised analysis would be performed and used.

"there is a lack of user-friendly methods for computational analysis of the LH data available to clinicians" The current manuscript does not provide enough detail to demonstrate clinician user-friendliness of the distributed code.

Minor comments:

add chmod 755 hormoneBayes (or similar) to the user guide

settings_example.csv (in the user guide) is settings_example.txt (in the distribution)

I get the following error when using the data and settings given in the user guide (mypath is shorthand for my path to the directory containg the executable):

Exception in Tkinter callback

Traceback (most recent call last):

File "tkinter/__init__.py", line 1892, in __call__

File "pulsatile_GUI.py", line 340, in start_analysis

File "shutil.py", line 264, in copyfile

FileNotFoundError: [Errno 2] No such file or directory: '/mypath/data/data.csv'

"hormoneBayes allows the input of two LH profiling datasets, corresponding to control and perturbed conditions" How is this done? The interface allows only one dataset.

Reviewer #2: The paper by Voliotis, et al. reports the development of HormoneBayes, a Bayesian model to identify pulsatile hormone dynamics from clinical measurements. The paper would benefit from the following suggestions.

1. Details about the testing of the method are not provided. How was the model applied to the different patients data? Was it the same model or was it different (i.e. different priors) for each type?

2. Details about the gold standard model are not present? What was the method used? How does it differ from HormoneBayes? What are the advantages of HormoneBayes over the existing method – that is, why would someone want to use HormoneBayes and not just use the ‘gold standard’ method? Confidence for the measurement is mentioned as an advantage, but more discussion needs to be added to detail what the differences are and the advantages of HormoneBayes.

3. HormoneBayes seems to produce some false negatives and some false positive results relative to the gold standard methods. The implications of this are not discussed. How impactful would these be on patient care and/or diagnosis?

4. If possible the gold standard method should be run on the larger dataset and results compared with those of HormoneBayes since this would add to the information about how it performs relative to the standard methods.

5. The discussion is somewhat limited and should include a recap of the main advantages of HormoneBayes (see point 2), the implications for HormoneBayes – what will it allow that is not currently possible?, a discussion of the practical aspects of HormoneBayes – what is the path for making it to actual use in the clinic?

6. A description of the software is lacking. That is, what language was it written in, what requirements (libraries, etc.) does it have, how does the GUI work, what system(s) is it available on, how easy would it be for someone with limited computational experience to install and use?

7. The Github repository listed appears to have the code available, and the use of a Docker container is welcome, but the documentation is very limited and should be part of the README, not just a Word document. Also, is the software limited to Macintosh only? That seems to be indicated in the Word document.

8. The data described in the paper and on which the results are based does not appear to be available anywhere and should be made available in a suitable fashion (potentially as part of the Github repository) so that the results could be replicated.

**Have the authors made all data and (if applicable) computational code underlying the findings in their manuscript fully available?**

Reviewer #1: None

Reviewer #2: **No: **Code is available, but it appears that data from the paper is not.

PLOS authors have the option to publish the peer review history of their article (what does this mean?). If published, this will include your full peer review and any attached files.

Reviewer #1: **Yes: **Tom Kelsey

Reviewer #2: No
---

## [Decision Letter · Decision Letter 2]

19 Feb 2024

Dear Dr Voliotis,

We are pleased to inform you that your manuscript 'HormoneBayes: a novel Bayesian framework for the analysis of pulsatile hormone dynamics' has been provisionally accepted for publication in PLOS Computational Biology.

Best regards,

Marc R Birtwistle, PhD

Academic Editor

PLOS Computational Biology

Virginia Pitzer

Section Editor

PLOS Computational Biology

Reviewer's Responses to Questions

**Comments to the Authors:**

Reviewer #1: My concerns at the R1 stage have been fully addressed in this version.

Reviewer #2: The authors have addressed all my comments and the manuscript is greatly improved.

**Have the authors made all data and (if applicable) computational code underlying the findings in their manuscript fully available?**

Reviewer #1: Yes

Reviewer #2: Yes

PLOS authors have the option to publish the peer review history of their article (what does this mean?). If published, this will include your full peer review and any attached files.

Reviewer #1: **Yes: **Tom Kelsey

Reviewer #2: No

---

## [Editor Report · Acceptance letter]

26 Feb 2024

PCOMPBIOL-D-23-00942R2 

HormoneBayes: a novel Bayesian framework for the analysis of pulsatile hormone dynamics

Dear Dr Voliotis,

I am pleased to inform you that your manuscript has been formally accepted for publication in PLOS Computational Biology. Your manuscript is now with our production department and you will be notified of the publication date in due course.

With kind regards,

Zsofia Freund
